# Comparison of Kinetic Models Applied for Transport Description in Polymer Inclusion Membranes

**DOI:** 10.3390/membranes13020236

**Published:** 2023-02-16

**Authors:** Piotr Szczepański

**Affiliations:** Faculty of Chemistry, Nicolaus Copernicus University, 7 Gagarina Street, 87-100 Toruń, Poland; piotrsz@umk.pl

**Keywords:** polymer inclusion membranes, kinetic models, model selection

## Abstract

Five mathematical models for transport description in polymer inclusion membranes (PIMs) were presented and compared via regression analysis. The applicability of the models was estimated through the examination of experimental data of Zn(II), Cd(II), Pb(II), and Cu(II) ions transported by typical carriers. In four kinetic models, a change in the feed and stripping solution volume was taken into account. The goodness of fit was compared using the standard error of the regression, Akaike information criterion (AIC), Bayesian (Schwarz) information criterion (BIC), and Hannan–Quinn information criterion (HQC). The randomness distribution in the data was confirmed via a nonparametric runs test. Based on these quantities, appropriate models were selected.

## 1. Introduction

In past decades, polymer inclusion membranes (PIMs) have been frequently used in analytical applications. PIMs are employed to separate and preconcentrate various species such as metals, organics, and pharmaceutical compounds [1,2,3,4,5]. Their use in analytics requires the development of quantitative descriptions of the transport kinetics, which can be obtained through mathematical modeling. Appropriate equations can be used for the calculation of time-dependent values of, for example, the selectivity, concentration, or recovery factors, since an accurate prediction of these quantities is important in analytics, especially in the procedures of sample preparation [6].

For many years, simple mathematical models similar to those describing the reaction kinetics have been used to describe transport in many membrane systems. These equations have been primarily applied for describing transport in liquid membrane (LM) and PIM systems. As their main advantage, they enable the calculation of the parameters that influence transport efficiency (e.g., maximum flux) and predict concentration changes in respective solutions of the adjacent membrane solutions.

### 1.1. Model No. 1

Despite the availability of many different kinetic equations, one of the most commonly employed mathematical models is that proposed by Danesi [7], whose mathematical form is identical to the differential equations describing the kinetics of first-order reactions:(1)$$\frac{dc_{f}}{dt}=-k_{1}c_{f}$$
(2)$$\frac{dc_{s}}{dt}=\frac{V_{f}}{V_{s}}k_{1}c_{f}$$
and results from the general transport scheme in the form of
(3)$$c_{f}\overset{k_{1}}{⟶}c_{s}$$
where *k*_1_ denotes the apparent first-order constant for membrane entrance, *V_f_* and *V_s_* are the volumes of the feed and receiving phase, and *c_f_* and *c_s_* are the concentrations of a substance at the given time *t* in the feed and the receiving solution, respectively.

The calculated *k*_1_ values can be further used for the evaluation of the permeability coefficient (*P*_1_, [cm/s]):(4)$$P_{1}=k_{1}\frac{V_{f}}{A}$$
as well as the initial maximum flux (*J_M_*, [mol/cm^2^·s]), because for the initial condition *t* = 0 and *c_f_* = *c_f,t_ =* 0, it can be estimated from the well-known relationship [7]
(5)$$J_{M}=-\frac{V_{f}}{A}\frac{dc_{f}}{dt}=P_{1}c_{f,t=0}$$

In the above equations, *A* denotes the membrane surface area [cm^2^]. This simple model, as represented by Equations (1) and (2), is most frequently used to describe transport kinetics in PIMs. For example, only last year, it was used to describe the removal of fluoride [8] and phenol [9], the separation of mercury(II) [10], the separation of lithium and magnesium [11,12], the recovery of bismuth(III) [13] and scandium [14], the extraction of arsenic(V) [15], the separation of Pb(II), Zn(II), and Cd(II) ions [16], and the removal of antibiotics [17].

Noteworthy is the fact that, in some cases, the model fit quality is poor [15,17] or does not satisfy the criterion of a random distribution of residuals [10,16]. This means that a different or more advanced model should be used to describe the transport kinetics.

### 1.2. Model No. 2

It was previously shown that the transport of Cd(II), Zn(II), Pb(II), and Cu(II) ions through PIMs containing reactive ionic liquids or D2EHPA (di-(2-ethylhexyl)phosphoric acid) as a carrier obeys the kinetic laws of a reversible first-order reaction [18,19]:(6)$$c_{f}\overset{k_{1}}{\underset{k_{-1}}{⇄}}c_{s}$$

According to the general transport scheme above, the rate of change in concentrations in the respective solutions during the membrane transport can be expressed by
(7)$$\frac{dc_{f}}{dt}=-k_{1}c_{f}+\frac{V_{s}}{V_{f}}k_{-1}c_{s}$$
(8)$$\frac{dc_{s}}{dt}=\frac{V_{f}}{V_{s}}k_{1}c_{f}-k_{-1}c_{s}$$

This model was also successfully applied for the description of Pb(II) ion transport through PIMs containing calixresorcin [4] arene derivatives as carriers [20].

### 1.3. Model No. 3

In the models presented above, the initial maximum flux can be interpreted as a flux related to the sorption of molecules into the membrane. In this case, the value of the initial flux into the receiving phase has no physical meaning because in membrane processes controlled by diffusion, a time lag should be expected. This means that the minimum period of time is needed for the formation of the complex and its penetration into the membrane interior until the release of the first molecules into the receiving phase. At the beginning of the process, the zero value of the transported substance flux into the receiving solution should be expected if the membrane has not been pre-equilibrated with the feed solution. The application of effective carriers and the appropriate membrane composition allow for minimizing the time lag effect to such an extent that it is not observable in the process. However, in order to describe transport through PIMs, considering the diffusive nature of the process, the time lag should be taken into account, as it is always present in this type of system. For this purpose, a model whose equations are similar to the equations describing the kinetics of first-order consecutive reactions can be used:(9)$$c_{f}\overset{k_{1}}{⟶}c_{LM}\overset{k_{2}}{⟶}c_{s}$$
(10)$$\frac{dc_{f}}{dt}=-k_{1}c_{f}$$
(11)$$\frac{dc_{LM}}{dt}=\frac{V_{f}}{V_{LM}}k_{1}c_{f}-k_{2}c_{LM}$$
(12)$$\frac{dc_{s}}{dt}=\frac{V_{LM}}{V_{s}}k_{2}c_{LM}$$

Such a mathematical model is mainly applied for the description of the transport of various substances through bulk liquid membranes (BLMs). For instance, this model was applied in the case of the pertraction of chromium(VI) [21,22], chromium(III) [23], chromium(II) [24], mercury(II) [25], cobalt(II) [26], cadmium (II) [27], and cyanide and thiocyanate ions [28,29].

### 1.4. Model No. 4

Analogous to the model proposed by Danesi [7], in this model, a problem with its fitting to experimental data, as well as the related incorrect residual distribution, is observed [21,23,28]. The quality of the model fit can be improved by applying a model similar to a consecutive reaction scheme with a slow reversible step, antecedent to the irreversible step, according to the scheme
(13)$$c_{f}\overset{k_{1}}{\underset{k_{-1}}{⇄}}c_{LM}\overset{k_{2}}{⟶}c_{s}$$
which can be described by the following set of differential equations:(14)$$\frac{dc_{f}}{dt}=-k_{1}c_{f}+\frac{V_{LM}}{V_{f}}k_{-1}c_{LM}$$
(15)$$\frac{dc_{LM}}{dt}=\frac{V_{f}}{V_{LM}}k_{1}c_{f}-k_{-1}c_{LM}-k_{2}c_{LM}$$
(16)$$\frac{dc_{s}}{dt}=\frac{V_{LM}}{V_{s}}k_{2}c_{LM}$$

This model was previously applied for the description of phenol [30], L-isoleucine [31], and strontium(II) pertraction [32] through BLMs.

In summary, there are at least four simple kinetic models that can be used to describe transport through PIMs. Nevertheless, only the model proposed by Danesi [7] is most frequently applied.

### 1.5. Model No. 5

It should be noted that in the case of the linear concentration vs. time dependence for transported substances, an equation similar to those describing a zero-order reaction (which corresponds to stationary or pseudo-stationary conditions of the transport) can be used:(17)$$\frac{dc_{f}}{dt}=-k_{1}$$

Despite the many different kinetic models that enable the determination of the initial maximum flux (*J_M_*) and a quantitative description of the concentration change in the feed and receiving solutions, some authors calculate *J_M_* from the first derivative of an exponential decay function fitted to the feed solution concentration dependence [33,34,35,36,37]. This approach probably results from problems occurring when fitting the most commonly used kinetic model (model No. 2) to the experimental data. The calculated *J_M_* value describes the system efficiency. However, it is impossible to obtain a quantitative description of time-dependent concentrations, especially in the receiving solution. In this case, there is also no possibility of a physicochemical interpretation of the exponential decay function, contrary to the typical kinetic models. The main aim of this work is, therefore, to discuss the applicability of the above simple kinetic models and to select the most appropriate model for the transport description of substances through PIMs. The usefulness of the models was evaluated with the use of typical parameters describing the quality of the model fit to the experimental results of Zn(II), Cd(II), Pb(II), and Cu(II) ion transport. In the model calculations, a change in the feed and stripping solution volume (because of sampling) was taken into account. The goodness-of-fit evaluation was estimated using the standard error of the regression, Akaike information criterion (AIC), Bayesian (Schwarz) information criterion (BIC), and Hannan–Quinn information criterion (HQC). A nonparametric runs test was also used to examine the randomness of the residuals.

The novelty of this research is the development of a method for the selection of a proper model based on appropriate fit quality parameters and a runs test as well as the application of kinetic models that have never been used for the description of transport through PIMs.

## 2. Experimental

Experimental studies of Zn, Cd, Pb, and Cu ion transport through PIMs containing various types of carriers were carried out. The following carriers were used for this purpose: TOPO (tri-n-octyl phosphine oxide, 90% Sigma Aldrich, St. Louis, MO, USA), D2EHPA (di-(2-ethylhexyl) phosphoric acid, 97% Aldrich, St. Louis, USA), Aliquat 336 (methyl trioctyl ammonium chloride, Aldrich St. Louis, MO, USA), Cyphos IL 101 (trihexyl(tetradecyl)phosphonium chloride, >97% Solvionic, Toulouse, France), and RILC8_Br (3-(1,3-diethoxy-1,3-dioxopropan-2-yl)-1-octylimidazolium bromide, synthesis described in [18]). Experimental studies were carried out in the system described in detail in [18,19]. The feed solution was composed of Zn(II), Cd(II), Cu(II), and Pb(II) nitrates (Sigma-Aldrich, St. Louis, MO, USA, reagent grade, purity ≥ 98%) dissolved in 0.5 M HCl (200 cm^3^) with an initial concentration equal to 0.002 M. Only in the system with D2EHPA as a carrier, due to its properties, was a 0.002 M solution of metal ions (initial pH = 4) without HCl used. As the stripping phase, a nitric acid solution with a concentration of 0.5 M and volume of 100 cm^3^ was applied. The surface membrane area was equal to 17 cm^2^. The aqueous solutions were pumped from the external reservoirs by a peristaltic pump (GILSON MINIPULS 3) at a 16 cm^3^/min flow rate. During the transport experiments, 1 mL samples from the feed and the receiving solution were taken for analysis with the flame atomic absorption method using a SPECTRAA 20ABQ Varian spectrophotometer. The atomic absorption spectroscopy operating parameters are presented in the Appendix A. All the experiments were carried out at room temperature (25 ± 2 °C).

### 2.1. Membrane Preparation

PIMs were prepared according to the procedure described in [18,19] using a solution casting and solvent evaporation technique. Cellulose triacetate (Acros Organics, Morris Plains, NJ, USA), as a polymer matrix, and o-nitrophenyl octyl ether (Alfa Aesar, Kandel, Germany, 98%), as a plasticizer, were used. The composition and thickness of the membranes applied in the experiments are presented in Table 1.

### 2.2. Model Calculations

The ordinary differential equations (ODEs) were solved with the Berkeley Madonna program v.8.1 (Berkeley, CA, USA) using the Rosenbrock (stiff) method. This method is related to the Runge–Kutta method; however, it possesses excellent stability properties, is computationally efficient, and preserves the positivity of the solutions [38,39]. The parameter values were estimated using the curve fit procedure which minimizes the deviation between the model output and dataset. The dependences of concentration changes in the feed and stripping solution were fitted simultaneously.

The most common parameter used to select the model with the best fit quality is the determination coefficient. However, it was proven that for nonlinear models, the application of the determination coefficient for the model selection is questionable because R-squared does not distinguish between good and bad nonlinear models [40].

Therefore, as the fit quality parameter, the standard error of the regression (*s_y_*) was calculated using the following equation:(18)$$s_{y}=\sqrt{MSE}=\sqrt{\frac{RSS}{n-p-1}}=\sqrt{\frac{\sum {\left(y_{i}-{\hat{y}}_{i}\right)}^{2}}{n-p-1}}$$
where *RSS* denotes the residual sum of squares, *MSE* is the mean squared error, *y_i_* is the observed value of the response variable, ${\hat{y}}_{i}$ is a predicted value of the response variable, *n* is the number of observations (the sample size), and *p* is the (total) number of estimated parameters.

Moreover, the criteria for model selection among a finite set of models, such as the Akaike information criterion (AIC), Bayesian (Schwarz) information criterion (BIC), and Hannan–Quinn information criterion (HQC), were also calculated. These criteria are most frequently applied as measures of the goodness of fit of a statistical model and are defined by [40]
(19)AIC=2⋅p - ln(L)
(20)BIC=p⋅ln(n) - 2⋅ln(L)
where ln(*L*) denotes the log-likelihood function of the statistical model defined by
(21)$$\ln(L)=0.5\cdot \left(-n\cdot \left(\ln(2\cdot \pi )+1-\ln n+\ln(RSS)\right)\right)$$

The Hannan–Quinn information criterion was calculated using the following equation [41]:(22)$$\mathrm{HQC}=n\cdot \ln\left(\frac{RSS}{n}\right)+2\cdot p\cdot \ln(\ln(n))$$

Note that the model with lower *s_y_*, AIC, BIC, and HQC values is preferred.

Before the model selection, another test should be performed to examine the random distribution of the residuals. For this purpose, a nonparametric runs test was used [42]. The best model was therefore selected from those in which the randomness of the residuals was fulfilled.

## 3. Results

### 3.1. Influence of the Feed and Stripping Solution Volume Changes

During the operation of the membrane system, samples of the feed and receiving solution were taken for concentration analysis (without return). Consequently, the effect of the change in the volume of the respective solutions was taken into account in the model calculations. The actual change in volume can be described by a step function, which is problematic to include in numerical calculations carried out in the Berkeley Madonna ODE solver. Therefore, a nonlinear function (third-degree polynomial) was used to describe the continuous change in volume during the system operation. As an example, in Figure 1a, the results of the calculations for the model proposed by Danesi are compared with those of the same model taking into account the influence of the feed and receiving solution volume (step and continuous). A comparison of the fits of the models represented by Equations (7) and (8) is presented in Figure 1b.

The results of the concentration vs. time dependence calculations for the feed solution are almost the same regardless of whether the change in volume is taken into account or not. For the stripping solution dependence, the application of the continuous (solid line) or step function (dotted line) for the description of the volume changes leads to a much better fit of the model to the experimental data. The percentage error between the calculated values of concentration in the receiving solution for the two best-fit models does not exceed 1%. Therefore, the solution volume changes in the membrane system were described by the third-degree polynomial in further model calculations. An example of a third-degree polynomial fitting to the experimental results is presented in the Appendix A.

### 3.2. TOPO as a Carrier

The experimental results of Cd(II), Zn(II), Pb(II), and Cu(II) ion transport through PIMs with TOPO as a carrier were used for the best model selection. No transport of Cu(II) ions was observed in this study, i.e., the concentration was below the detection limit of AAS. The values of the fit quality parameters and the initial maximum fluxes are listed in Table 2. The lowest values are in bold.

The results indicate that in the case of Cd(II) and Zn(II) ion transport, the lowest values of the *s_y_*, AIC, BIC, and HQC parameters were obtained for model No. 4, while model No. 3 was best-fitted to the experimental data of Pb(II) ion transport. Despite the similarity of the values of the initial fluxes calculated using different models, it should be remembered that the fit quality of the models is different. For example, Figure 2 depicts the fit of model No. 1 (dashed line) and model No. 4 (solid line) to the experimental data of Zn(II) transport. Model No. 1 was the worst-fit model, whereas model No. 4 was the best-fit model. Furthermore, model No. 1 (and models No. 2 and No. 3) also failed the runs test because the residuals were not randomly distributed. The best-fit models for all the transported ions and systems are presented in the Appendix A.

The Zn(II) concentration differences in the feed solution, calculated by models No. 1 and No. 4, were small. Therefore, the same values of the initial maximum fluxes (*J_M_* = 3.400 × 10^−10^ mol/cm^2^·s) were evaluated. Significant differences are visible only in the concentration dependences for the receiving solution, leading to a maximum percentage error exceeding 5% at the end of the transport. Such a value is inacceptable when the model is used in analytical applications, e.g., to predict the enrichment factor of analytes.

### 3.3. Aliquat 336 as a Carrier

The kinetic models were applied for the description of the transport of Cd(II), Zn(II), Pb(II), and Cu(II) ions through PIMs containing Aliquat 336 as a carrier. Similar to the system with TOPO as a carrier, transport of Cu(II) ions was not observed (below the detection limit of AAS). The calculated model fit quality parameters are presented in Table 3, along with the initial maximum fluxes. Based on these quantities, the most appropriate models were selected.

Model No. 4 provided the lowest values of all the model fit quality parameters for Zn(II) and Pb(II) ion transport, whereas model No. 3 showed the best fit in the case of Cd(II) ion transport. In this membrane system, the best-fitted models led to initial maximum fluxes higher than 7 to 11%, especially for preferentially transported Cd(II) and Zn(II) ions.

A comparison of the fits of models No. 3 and No. 1 to the experimental data for Cd(II) ion transport is shown in Figure 3.

The experimental results and the fitted model No. 3 indicate the occurrence of a time lag in the investigated system and thus the accumulation of transported Cd(II) in the membrane. The same effect was also observed for Zn(II) and Pb(II) ion transport. This phenomenon is fundamental in analytical applications since accumulation reduces the transport efficiency and enrichment factor. Note that for models No. 1 and No. 2, the accumulation of the transported substance in the membrane was not taken into account.

### 3.4. Cyphos IL 101 as a Carrier

The fit quality measures obtained from fitting the kinetic models to the experimental data of Cd(II), Zn(II), Pb(II), and Cu(II) ion transport through PIMs containing Cyphos IL 101 as a carrier are presented in Table 4.

The lowest values—which indicate a better fit of the model to the experimental data—were observed for model No. 3 used for the description of the Cd(II) and Pb(II) concentration vs. time dependences. In the case of Zn(II) and Cu(II) ion transport, only one model satisfied the criterion of the randomness of the data. For the Cu(II) ions, it was model No. 2, and for the Zn(II) ions, it was model No. 4. The results show that, similar to the system with Aliquat 336 as the carrier, for Cd(II), Zn(II), and Pb(II) ion transport, accumulation in the membrane was also observed. The biggest difference between the initial maximum flux values calculated from the appropriate models was found for the transport of Cu(II) ions; the *J_M_* calculated by the correctly fitted model was over 55% higher than the values calculated by the other models. The fitting of the worst (model No. 1) and best models (No. 4) to the experimental data of Zn(II) ion transport is compared in Figure 4.

The fits of models No. 1 and No. 4 to the dependence of the concentration changes in the feed solution were similar, leading to almost identical flux values of Zn(II) ions. For Cyphos IL 101 as a carrier in PIMs, accumulation of transported Cd(II), Zn(II), and Pb(II) ions in the membrane was also observed. Differences in the fit of the models appear for the receiving solution concentration vs. time dependence, indicating that the fit of model No. 5 is much better.

### 3.5. D2EHPA as a Carrier

The calculated fit quality characteristics of the analyzed models used for the description of Cd(II), Zn(II), Pb(II), and Cu(II) ion transport through PIMs containing D2EHPA as a carrier are compared in Table 5.

The transport kinetics in this system can be described by various models satisfying the criterion of the randomness of the data, especially for the transport of Cd(II) ions. From the comparison of the fit quality parameter results, all of the fit criteria using model No. 3 were lower than those of the other models for Cd(II) and Zn(II) ion transport. The lowest *s_y_*, AIC, BIC, and HQC values for Pb(II) ion transport were found for model No. 4, while for Cu(II) ions, the lowest values were found for model No. 2. The calculated initial maximum fluxes using different models were similar. However, a decidedly different fit of the models to the experimental data was observed (see Figure 5). For Cu(II) ion transport, the initial maximum flux calculated by model No. 2 was approx. 14% higher when compared to the values calculated by the other models.

### 3.6. Reactive Ionic Liquid (RILC8_Br) as a Carrier

A summary of the fit quality measures of the analyzed models is presented in Table 6. In this membrane system, no transport of Cu(II) ions was observed.

All the fit quality measures indicate that there was only one kinetic model appropriate for the description of Cd(II), Zn(II), and Pb(II) ion transport through PIMs using RILC8_Br as a carrier. In the case of Cd(II) and Zn(II) ion transport, except for model No. 2, no other model met the random distribution of residuals criterion. Choosing an inappropriate model leads to substantial errors in the initial maximum flux values. The calculated *J_M_* values from the best-fit model were higher by approx. 22% for Cd(II) ions and up to 74% for Zn(II) ions. The proper choice of this model for the description of metal ion transport through PIMs containing reactive ionic liquids was also confirmed by the results presented in [18].

The fit of models No. 2 and No. 4 to the experimental data of Cd(II) ion transport is shown in Figure 6. The results indicate a much better fit of model No. 2. Moreover, no time lag effect was observed in the investigated system. The undetectable time lag may result primarily from the properties of the carrier used, as well as the application of the thinnest membrane in the experiments.

## 4. Conclusions

An appropriate mathematical model enables a quantitative prediction of concentration changes in external solutions and the calculation of, e.g., the time needed to achieve the maximum concentration, recovery, or enrichment factor of the transported substances.

Among the numerous equations used to describe transport kinetics, it is impossible to choose only one as the most appropriate. The presented results indicated that the application of the most frequently used model (proposed by Danesi) is significantly limited because of the nonrandomly distributed residuals.

The selection of the appropriate model should be based on the values of the standard error of the regression, Akaike information criterion (AIC), Bayesian (Schwarz) information criterion (BIC), or Hannan–Quinn information criterion (HQC) after the runs test evaluation (residuals’ randomness check).

Of the models presented in this report, model No. 4 is the most universal. However, the model selection should be individualized for each experimental relationship. It was also found that a nonlinear equation (third-degree polynomial) can be successfully used to describe solutions’ volume changes in a membrane system (because of sampling), leading to a better fit of the model to the experimental data.

The results indicated that the models that have not been used thus far for transport description in PIMs, i.e., models No. 3 and No. 4, can be successfully applied. These models are particularly important in the case of systems where a time lag is observed. This mainly applies to systems with carriers characterized by a high partition coefficient (high sorption of transported substances to the membrane), slow diffusion inside the membrane (e.g., because of the relatively high membrane thickness), or slow kinetics of extraction and re-extraction at the respective membrane interfaces.

## Figures and Tables

**Figure 1 membranes-13-00236-f001:**
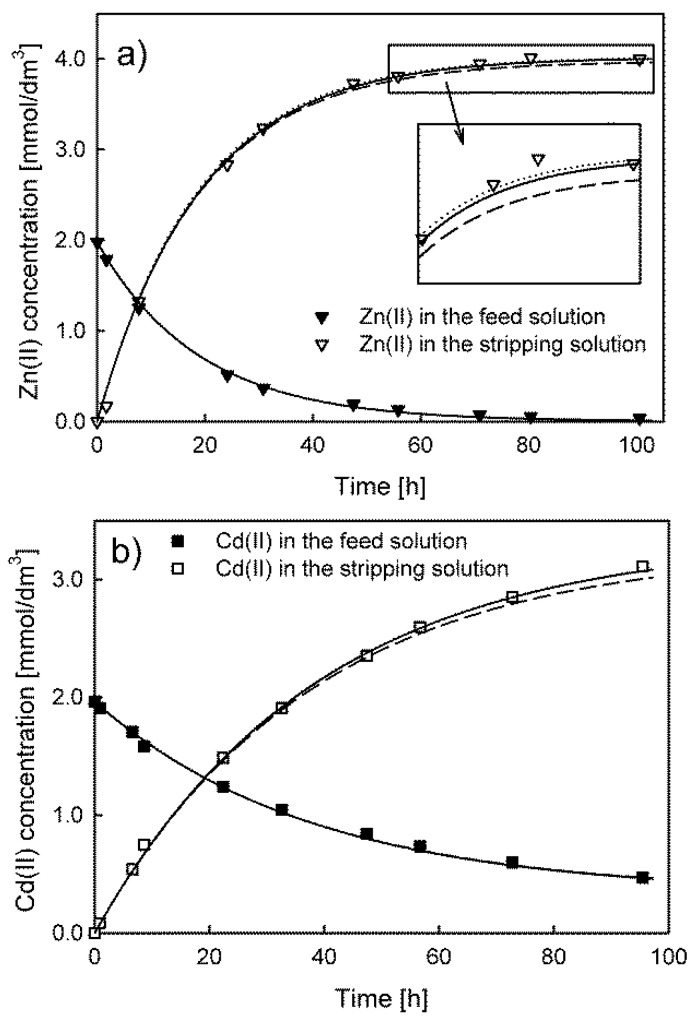
Comparison of the model that does not take into account the volume changes (dashed lines) with the models using a continuous function (solid lines) and a step function (dotted lines) to describe the changes in the volume. The systems use D2EHPA (**a**) and RILC8_Br (**b**) as carriers.

**Figure 2 membranes-13-00236-f002:**
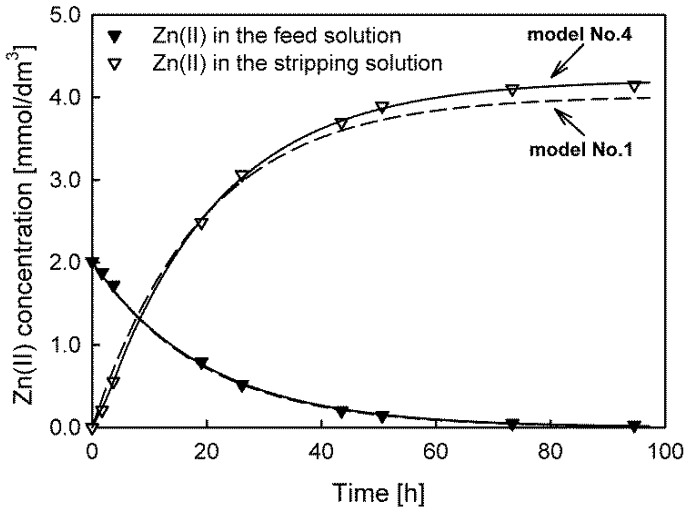
Comparison of the best (model No. 4, solid line) and worst (model No. 1, dashed line) model fits for predicting Zn(II) ion transport through PIMs with TOPO as a carrier.

**Figure 3 membranes-13-00236-f003:**
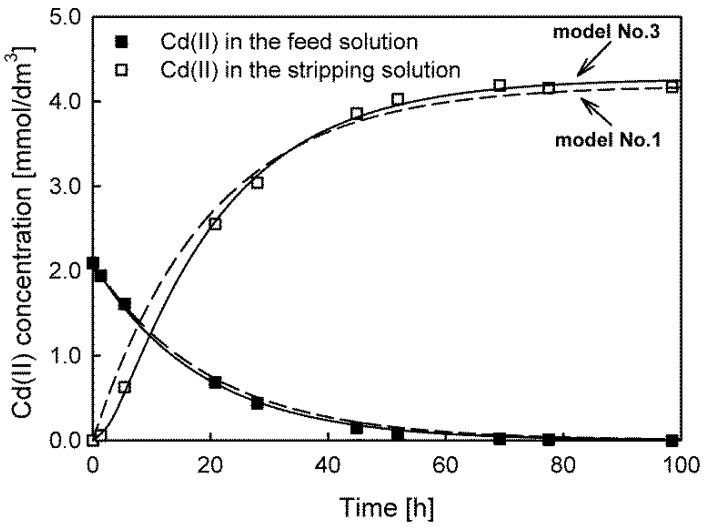
The best (model No. 3, solid line) and worst (model No. 1, dashed line) model fits for predicting Cd(II) ion transport through PIMs with Aliquat 336 as a carrier.

**Figure 4 membranes-13-00236-f004:**
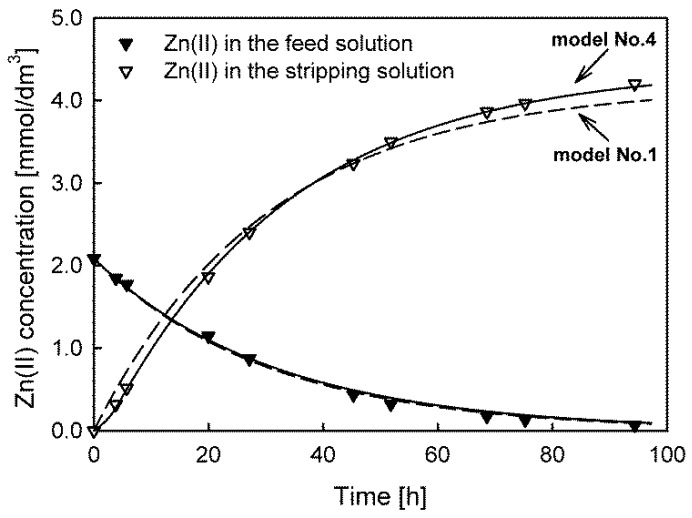
The best (model No. 4, solid line) and worst (model No. 1, dashed line) model fits for predicting Zn(II) ion transport through PIMs with Cyphos IL 101 as a carrier.

**Figure 5 membranes-13-00236-f005:**
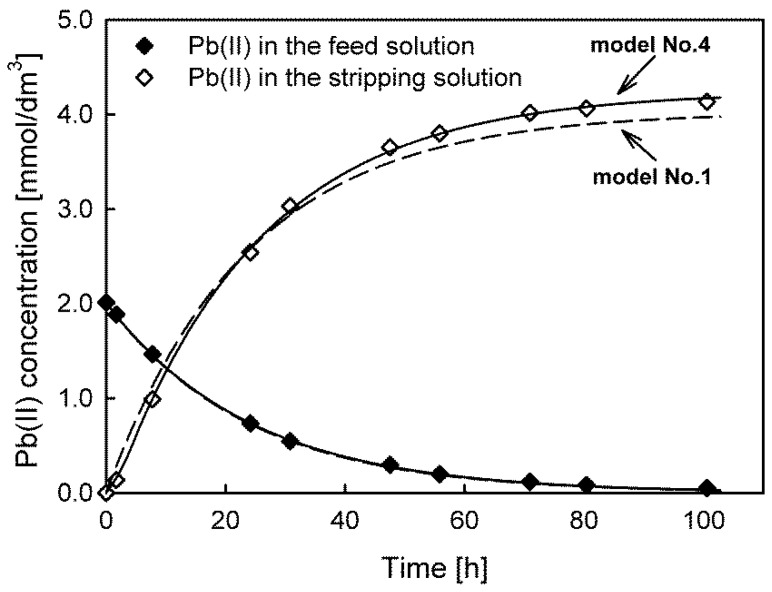
Comparison of the best (model No. 4, solid line) and worst (model No. 1, dashed line) model fits for predicting Pb(II) ion transport through PIMs with D2EHPA as a carrier.

**Figure 6 membranes-13-00236-f006:**
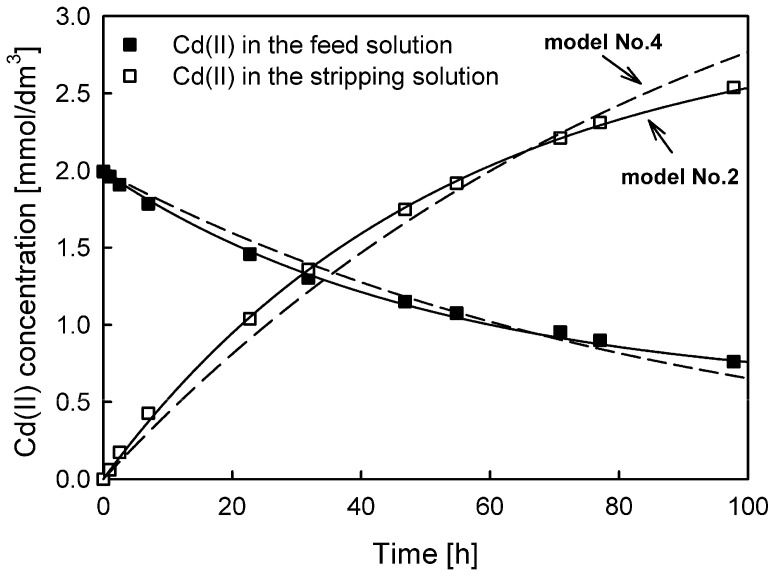
Comparison of the best model fit (model No. 2, solid line) and the fit of the model represented by Equations (14)–(16) (model No. 4, dashed line) to the experimental data of Cd(II) ion transport through PIMs with RILC8_Br as a carrier.

**Table 1 membranes-13-00236-t001:** The composition and thickness of the membranes used.

Carrier Type	Membrane Composition [wt%]			Thickness [mm]
	Carrier	NPOE	CTA	
TOPO	52	30	18	0.100 (±0.005)
Aliquat 336	44	31	25	0.149 (±0.011)
Cyphos IL 101	49	19	32	0.1498 (±0.0014)
D2EHPA	46	29	25	0.199 (±0.043)
RILC8_Br	50	20	30	0.0972 (±0.0074)

**Table 2 membranes-13-00236-t002:** The calculated fit quality parameters and initial maximum fluxes for the system with TOPO as a carrier.

Ion	No.	Scheme	*V* = f(*t*)	Randomness in Data (Runs Test)	*s_y_* × 10^5^	AIC	BIC	HQC	*J_M_* × 10^10^ [mol/cm^2^·s]
**Cd (II)**	1	A→B	NO	NO	7.97	−288.76	−287.87	−339.72	3.006
	1a	A→B	YES	NO	6.49	−296.16	−295.27	−347.12	2.926
	2	A↔B	YES	NO	6.70	−294.18	−292.40	−345.01	2.924
	3	A→B→C	YES	YES	3.67	−315.82	−311.15	−366.66	3.036
	4	A↔B→C	YES	YES	**3.21**	**−319.94**	**−317.27**	**−370.65**	2.989
**Zn (II)**	1	A→B	NO	NO	9.66	−281.85	−280.96	−332.81	3.400
	1a	A→B	YES	NO	7.52	−290.86	−289.97	−341.82	3.332
	2	A↔B	YES	NO	7.77	−288.86	−287.08	−339.69	3.332
	3	A→B→C	YES	NO	5.57	−300.85	−296.18	−351.69	3.477
	4	A↔B→C	YES	YES	**2.87**	**−324.00**	**−321.33**	**−374.71**	3.400
**Pb (II)**	1	A→B	NO	YES	1.68	−344.75	−343.86	−395.71	0.2411
	1a	A→B	YES	YES	1.52	−348.50	−347.61	−399.46	0.2342
	2	A↔B	YES	YES	**1.26**	**−354.47**	**−352.69**	**−405.31**	0.2485
	3	A→B→C	YES	YES	1.57	−346.50	−341.83	−397.33	0.2342
	4	A↔B→C	YES	YES	1.61	−344.82	−342.15	−395.54	0.2342

**Table 3 membranes-13-00236-t003:** The calculated fit quality parameters and initial maximum fluxes for the system with Aliquat 336 as a carrier.

Ion	No.	Scheme	*V* = f(*t*)	Randomness in Data (Runs Test)	*S_y_* × 10^5^	AIC	BIC	HQC	*J_M_* × 10^10^ [mol/cm^2^·s]
**Cd (II)**	1	A→B	NO	NO	13.1	−301.06	−300.07	−357.62	3.476
	1a	A→B	YES	YES	12.2	−303.68	−302.69	−360.25	3.347
	2	A↔B	YES	YES	12.6	−301.71	−302.72	−358.08	3.345
	3	A→B→C	YES	YES	**4.53**	**−342.57**	**−343.57**	**−398.94**	3.718
	4	A↔B→C	YES	YES	4.67	−340.57	−343.57	−396.74	3.718
**Zn (II)**	1	A→B	NO	NO	14.8	−295.99	−294.99	−352.55	1.512
	1a	A→B	YES	NO	13.6	−299.52	−298.52	−356.08	1.473
	2	A↔B	YES	NO	14.0	−297.52	−298.52	−353.89	1.473
	3	A→B→C	YES	YES	8.59	−316.98	−317.99	−373.35	1.680
	4	A↔B→C	YES	YES	**6.36**	**−328.20**	**−331.21**	**−384.38**	1.620
**Pb (II)**	1	A→B	NO	NO	3.06	−359.07	−358.07	−415.63	0.3340
	1a	A→B	YES	YES	2.82	−362.46	−361.46	−419.02	0.3265
	2	A↔B	YES	YES	2.90	−360.46	−361.46	−416.83	0.3265
	3	A→B→C	YES	YES	2.18	−371.92	−372.92	−428.29	0.3363
	4	A↔B→C	YES	YES	**2.01**	**−374.30**	**−377.30**	**−430.47**	0.3424

**Table 4 membranes-13-00236-t004:** The calculated fit quality parameters and initial maximum fluxes for the system with Cyphos IL 101 as a carrier.

Ion	No.	Scheme	*V* = f(*t*)	Randomness in Data (Runs Test)	*s_y_* × 10^5^	AIC	BIC	HQC	*J_M_* × 10^10^ [mol/cm^2^·s]
**Cd (II)**	1	A→B	NO	NO	9.53	−313.70	−312.70	−370.26	3.206
	1a	A→B	YES	NO	8.96	−316.14	−315.15	−372.70	3.067
	2	A↔B	YES	YES	9.22	−314.15	−312.15	−370.51	3.067
	3	A→B→C	YES	YES	**3.30**	**−355.19**	**−350.21**	**−411.56**	3.240
	4	A↔B→C	YES	YES	3.41	−353.20	−350.22	−409.38	3.241
**Zn (II)**	1	A→B	NO	NO	10.6	−309.47	−308.48	−366.03	2.254
	1a	A→B	YES	NO	8.95	−316.17	−315.18	−372.74	2.190
	2	A↔B	YES	NO	9.21	−314.17	−312.18	−372.73	2.190
	3	A→B→C	YES	NO	5.00	−338.60	−333.61	−397.16	2.307
	4	A↔B→C	YES	YES	**4.00**	**−346.78**	**−343.79**	**−407.34**	2.234
**Pb (II)**	1	A→B	NO	NO	26.9	−272.21	−271.21	−328.77	0.7189
	1a	A→B	YES	NO	3.79	−350.57	−349.57	−407.13	0.7000
	2	A↔B	YES	NO	3.90	−348.57	−346.57	−404.93	0.7000
	3	A→B→C	YES	YES	**3.03**	**−358.61**	**−353.63**	**−414.98**	0.7170
	4	A↔B→C	YES	YES	3.09	−356.98	−353.99	−413.15	0.7168
**Cu (II)**	1	A→B	NO	NO	0.312	−225.39	−225.09	−254.10	0.007139
	1a	A→B	YES	NO	0.322	−224.73	−224.42	−253.44	0.006916
	2	A↔B	YES	YES	**0.132**	**−241.96**	**−241.36**	**−271.01**	0.01110
	3	A→B→C	YES	NO	0.345	−222.73	−219.82	−251.77	0.006916
	4	A↔B→C	YES	NO	0.373	−220.73	−219.83	−250.11	0.006917

**Table 5 membranes-13-00236-t005:** The calculated fit quality parameters and initial maximum fluxes for the system with D2EHPA as a carrier.

Ion	No.	Scheme	*V* = f(*t*)	Randomness in Data (Runs Test)	*s_y_* × 10^5^	AIC	BIC	HQC	*J_M_* × 10^10^ [mol/cm^2^·s]
**Cd (II)**	1	A→B	NO	YES	4.10	−347.39	−346.39	−403.95	0.4684
	1a	A→B	YES	YES	4.29	−345.58	−344.58	−402.14	0.4548
	2	A↔B	YES	YES	4.43	−343.49	−341.50	−399.86	0.4579
	3	A→B→C	YES	YES	**2.90**	**−360.39**	**−355.41**	**−416.76**	0.4757
	4	A↔B→C	YES	YES	3.38	−353.53	−350.54	−409.71	0.4652
**Zn (II)**	1	A→B	NO	NO	6.36	−329.88	−328.89	−386.44	3.514
	1a	A→B	YES	NO	5.24	−337.63	−336.63	−394.19	3.413
	2	A↔B	YES	NO	5.42	−335.39	−333.40	−391.76	3.421
	3	A→B→C	YES	YES	**3.44**	**−353.59**	**−348.61**	**−409.96**	3.527
	4	A↔B→C	YES	YES	3.51	−352.03	−349.04	−408.21	3.512
**Pb (II)**	1	A→B	NO	NO	10.3	−310.67	−309.68	−367.24	2.792
	1a	A→B	YES	YES	7.91	−321.14	−320.15	−377.71	2.747
	2	A↔B	YES	YES	8.14	−319.14	−320.15	−375.51	2.747
	3	A→B→C	YES	NO	5.75	−333.02	−334.03	−389.39	2.881
	4	A↔B→C	YES	YES	**2.37**	**−367.68**	**−370.68**	**−423.85**	2.822
**Cu (II)**	1	A→B	NO	YES	2.27	−371.02	−370.02	−427.58	0.2180
	1a	A→B	YES	NO	2.29	−370.80	−369.81	−427.37	0.2121
	2	A↔B	YES	YES	**1.31**	**−392.16**	**−393.16**	**−448.53**	0.2425
	3	A→B→C	YES	NO	2.35	−368.80	−369.81	−425.17	0.2121
	4	A↔B→C	YES	NO	2.42	−366.88	−369.89	−423.06	0.2122

**Table 6 membranes-13-00236-t006:** The calculated fit quality parameters and initial maximum fluxes for the system with a reactive ionic liquid (RILC8_Br) as a carrier.

Ion	No.	Scheme	*V* = f(*t*)	Randomness in Data (Runs Test)	*s_y_* × 10^5^	AIC	BIC	HQC	*J_M_* × 10^11^ [mol/cm^2^·s]
**Cd (II)**	1	A→B	NO	NO	7.19	−322.84	−321.95	−379.41	7.499
	1a	A→B	YES	NO	7.04	−323.72	−322.82	−380.28	7.260
	2	A↔B	YES	YES	**2.286**	**−367.70**	**−365.93**	**−424.08**	9.190
	3	A→B→C	YES	NO	8.74	−314.07	−309.40	−370.44	7.260
	4	A↔B→C	YES	NO	9.00	−311.99	−309.32	−368.16	7.260
**Zn (II)**	1	A→B	NO	NO	4.99	−337.41	−336.52	−393.97	1.386
	1a	A→B	YES	NO	5.09	−336.61	−335.72	−393.17	1.345
	2	A↔B	YES	YES	**1.66**	**−380.57**	**−378.79**	**−436.95**	2.414
	3	A→B→C	YES	NO	5.23	−334.61	−329.94	−390.98	1.345
	4	A↔B→C	YES	NO	5.69	−332.61	−329.94	−388.78	1.346
**Pb (II)**	1	A→B	NO	YES	1.79	−378.51	−377.62	−435.07	1.075
	1a	A→B	YES	YES	1.84	−377.49	−376.59	−434.05	1.049
	2	A↔B	YES	YES	**0.651**	**−417.98**	**−416.19**	**−474.34**	1.344
	3	A→B→C	YES	YES	1.88	−375.49	−370.81	−431.86	1.049
	4	A↔B→C	YES	YES	1.93	−373.53	−370.86	−429.70	1.049

## Data Availability

Data is contained within the article or Appendix A.

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
