# Peer review of "Comparison of Kinetic Models Applied for Transport Description in Polymer Inclusion Membranes"

_membranes, 2023, doi:10.3390/membranes13020236_

Round 1
Reviewer 1 Report
Manuscript Title: Comparison of kinetic models applied for the transport description in polymer inclusion membranes
Recommendation: Minor Revision
The authors have to be appreciated for their contribution on application of mathematical modeling for the description transport through PIMs. In this work, five mathematical models applied for the transport description through polymer inclusion membranes (PIMs) were presented and compared. The usefulness of models was estimated by means of experimental results of Zn(II), Cd(II), Pb(II), and Cu(II) ions transport using typical carriers. In four kinetic models, a change in the feed and stripping solution volume was taken into account. The goodness-of-fit evaluation was carried out using the standard error of the regression, 14 Akaike information criterion (AIC), Bayesian (Schwarz) information criterion (BIC), and Hannan Quinn information criterion (HQC). Randomness distribution in data was confirmed by a nonparametric runs test. Based on these quantities, appropriate models were selected.
The manuscript is acceptable after the below mentioned corrections are being addressed by the authors.
1. It is important to label a) and b) for fig. 1, since, the authors have mentioned Fig. 1a in line no. 187, and Fig. 1b in line no 191.
2. It is better to add 1-2 sentences of information about the role of Rosenbrock method, with citing the relevant references.
3. In line no. 209, it is written as Tab., but in the rest of the cases it is written as Table.
4. What does square root of MSE denotes in eqn. 18?
5. The referencing needs to be improved, few relevant papers are provided below
https://doi.org/10.1016/j.chemosphere.2021.130604
https://doi.org/10.1016/j.cherd.2022.03.052
https://doi.org/10.1016/j.microc.2020.104980
https://doi.org/10.1016/j.memsci.2012.03.061
Author Response
I would like to thank the referee for evaluating my manuscript. I have tried to address all the reviewers’ concerns in a proper way and believe that my paper has improved considerably.
Rev#1
Recommendation: Minor Revision
The authors have to be appreciated for their contribution on application of mathematical modeling for the description transport through PIMs. In this work, five mathematical models applied for the transport description through polymer inclusion membranes (PIMs) were presented and compared. The usefulness of models was estimated by means of experimental results of Zn(II), Cd(II), Pb(II), and Cu(II) ions transport using typical carriers. In four kinetic models, a change in the feed and stripping solution volume was taken into account. The goodness-of-fit evaluation was carried out using the standard error of the regression, 14 Akaike information criterion (AIC), Bayesian (Schwarz) information criterion (BIC), and Hannan Quinn information criterion (HQC). Randomness distribution in data was confirmed by a nonparametric runs test. Based on these quantities, appropriate models were selected.
The manuscript is acceptable after the below mentioned corrections are being addressed by the authors.
Q1: It is important to label a) and b) for fig. 1, since, the authors have mentioned Fig. 1a in line no. 187, and Fig. 1b in line no 191.
RESPONSE: Label a) and b) are already added.
Q2: It is better to add 1-2 sentences of information about the role of Rosenbrock method, with citing the relevant references.
RESPONSE:
According to the Reviewer #1 expectations some additional explanations were added:
“...which is related to the Runge-Kutta methods, however possessing excellent stability properties, is computationally efficient and preserving positivity of the solutions [33, 34].
[33] Voss, D.A.; Khaliq, A.Q.M. Parallel Rosenbrock methods for chemical systems. Comput. Chem. 2001, 25, 101–107, DOI: 10.1016/S0097-8485(00)00093-0.
[34] Ostermann, A.; Roche, M. Rosenbrock methods for partial differential equations and fractional orders of convergence. SIAM J. Numer. Anal. 1993, 30(4), 1084-1098, Available online: http://www.jstor.org/stable/2158191 (accessed on 2 Jan. 2023).
“The parameter values were estimated using curve fit procedure, which minimize the deviation between model output and a dataset.“
Q3: In line no. 209, it is written as Tab., but in the rest of the cases it is written as Table.
RESPONSE: Done
Q4: What does square root of MSE denotes in eqn. 18?
RESPONSE: additional explanation was added:
MSE - the mean squared error,
Q5: The referencing needs to be improved, few relevant papers are provided below
https://doi.org/10.1016/j.chemosphere.2021.130604
https://doi.org/10.1016/j.cherd.2022.03.052
https://doi.org/10.1016/j.microc.2020.104980
https://doi.org/10.1016/j.memsci.2012.03.061
RESPONSE: Citations were added:
[4] Keskin, B.; Zeytuncu-Gökoğlu, B.; Koyuncu, I. Polymer inclusion membrane applications for transport of metal ions: A critical review. Chemosphere 2021, 279, 130604, DOI: 10.1016/j.chemosphere.2021.130604.
[5] López-Guerrero, M.M.; Granado-Castro, M.D.; Díaz-de-Alba, M.; Lande-Durán, J.; Casanueva-Marenco, M.J. A polymer inclusion membrane for the simultaneous determination of Cu(II), Ni(II) and Cd(II) ions from natural waters. Microchem. J. 2020, 157, 104980, DOI: 10.1016/j.microc.2020.104980.
Note that https://doi.org/10.1016/j.cherd.2022.03.052 was already provided in the manuscript.
In the case of paper:
St John, A.M.; Cattrall, R.W.; Kolev, S.D. Transport and separation of uranium(VI) by a polymer inclusion membrane based on di-(2-ethylhexyl) phosphoric acid. J. Membr. Sci. 2012, 409–410, 242-250, DOI: 10.1016/j.memsci.2012.03.061.
I decided not to include it in the manuscript because it is relatively old and does not concern the analytical applications of PIMs.
Reviewer 2 Report
Comments and suggestions
The summary is poorly written, it must include the objective of the study, the main method and the results.
- The introduction is poorly designed. It is better to send the detail of the equations as supplementary material.
- It is recommended to group the main equations in a table and to number them according to the order used in the manuscript.
- It is necessary to specify (indicate) the models on the graphs (for example: ---- model N.1)
- In the tables, the charge of the ion must be mentioned (eg Cd(II) instead of Cd).
- Adopt the same number of decimal places in parameter values throughout the manuscript.
- In a similar previous study published (https://doi.org/10.1016/j.memsci.2021.119674), the author reported a flux J = 2.60 (± 0.23) × 10-13 mol/cm2s for the Cu(II) ions, whereas in the submitted study the author indicates that the copper ions do not diffuse through the same previous membrane (J = 0).
- Major contradiction and self-plagiarism on the part of the author
- There is no agreement between the results obtained, the validated models and the physico-chemical transport phenomena since the validated model changes from one combination (ion-transporter) to another.
- The discussion and explanation of the results are insufficient and the interpretation is superficial, resembling a theoretical chemistry article.
Example: - Explain why you did not observe a diffusion of Cu(II) ions through the membrane with TOPO as carrier.
- Specify the operating conditions of the work: temperature, acidity, agitation, etc.
· There are no studies to characterize engineered membranes, (SEM/EDS/IFTR)
Author Response
I would like to thank the referee for evaluating my manuscript. I have tried to address all the reviewers’ concerns in a proper way and believe that my paper has improved considerably.
REV#2
Q1: The summary is poorly written, it must include the objective of the study, the main method and the results.
RESPONSE: I tried to take into account all of the expectations, and I changed conclusions to:
The primary purpose of this manuscript was to compare five simple models used for the kinetic transport description of Cd(II), Zn(II), Pb(II), and Cu(II) ions through PIMs containing various carriers. The best-fit model is characterized by the lowest model–fit quality parameter values (sy, AIC, BIC, and HQC) and a random distribution of residuals (checked by runs test). The presented results indicated that the application of the most frequently used model (proposed by Danesi) is significantly limited. Among the many equations used to describe transport kinetics, it is impossible to choose only one as the most appropriate. From the models presented in this manuscript, the most universal seems to be model No. 4, based on differential equations similar to those which describe consecutive reactions with a slow reversible step, antecedent to the irreversible step. However, the model selection should be individual for each experimental relationship.
It was also found that the non–linear equation (3rd–degree polynomial) can be successfully used to describe the solutions volume changes in a membrane system (because of sampling) and leads to a better fit of the model to the experimental data.
Selecting the inappropriate model leads to underestimation or overestimation of the initial maximum flux value and disallows quantitative prediction of concentration changes in external solutions, which is particularly important in analytical applications of PIMs, e.g., for calculation, for example, the time needed to achieve maximum concentration, recovery, or enrichment factor of transported substances
Q2: The introduction is poorly designed. It is better to send the detail of the equations as supplementary material.
RESPONSE: It is always hard to describe many various mathematical models (equations). Therefore I decided use only differential equations in order to describe the presented models. Omitting these equations in the introduction section would make it even more difficult to understand how a given model differs from others. In the reviewed version, I decided to add subsections to highlight particular models. The models were also renumbered (No.2 to No.1a, No.3 to No.2, etc.).
Q3: It is recommended to group the main equations in a table and to number them according to the order used in the manuscript.
RESPONSE: According to the Reviewer expectations, all differential equations are compared in the Supplementary file (Table S1.)
Q4: It is necessary to specify (indicate) the models on the graphs (for example: ---- model N.1)
RESPONSE: Done.
Q5: In the tables, the charge of the ion must be mentioned (eg Cd(II) instead of Cd).
RESPONSE: It was done in the first version of my manuscript. However, after editing, it disappeared (now fixed).
Q6: Adopt the same number of decimal places in parameter values throughout the manuscript.
RESPONSE: The same number of the significant figure is applied in my manuscript. For maximum initial fluxes, I used four significant figures. For fit quality parameters (AIC, BIC, and HQC) I applied five because, in some cases, these values differ on the first (or even second) decimal place. For the same reason, in the case of the standard error of the regression (sy), I applied three significant figures; however, this value should be rounded up to two significant figures.
Q7: In a similar previous study published (https://doi.org/10.1016/j.memsci.2021.119674), the author reported a flux J = 2.60 (± 0.23) × 10-13 mol/cm2s for the Cu(II) ions, whereas in the submitted study the author indicates that the copper ions do not diffuse through the same previous membrane (J = 0). Major contradiction and self-plagiarism on the part of the author
RESPONSE: First of all, the experimental results presented in this manuscript are new and have not been published anywhere! Hence the differences in flux values and observed transport of ions. Indeed in the case of Aliquat 336, and the membrane used in [16], a very low Cu(II) flux was observed (Cu concentration in the receiving solution slightly higher than the detection limit by AAS). Note that the concentration of carrier was higher (49 vs. 44wt%) as well as the thickness was lower (0.090 vs. 0.149 mm). Probably because of these differences, the Cu(II) ion transport was not observed. The same effect (no Cu(II) transport) was observed in the case of TOPO as a carrier.
To fulfill the Reviewer expectations, an explanation was added:
No transport of Cu(II) ions was observed in the conducted studies (concentration below the detection limit of AAS).
and:
“Similarly to the system with TOPO as a carrier, the transport of Cu(II) ions was not observed (below the detection limit of AAS).”
Q9: There is no agreement between the results obtained, the validated models and the physico-chemical transport phenomena since the validated model changes from one combination (ion-transporter) to another.
RESPONSE: There is no contradiction between the results and fitted models. The presented results clearly indicate that there is no one “the best” model (e.g., proposed by Danesi) which can be applied to all of the systems, concentrations, carriers, etc. The choice of the model should be made in the way proposed in this work (i.e., best fitted).
Q10: The discussion and explanation of the results are insufficient and the interpretation is superficial, resembling a theoretical chemistry article.
RESPONSE: The main aim of this manuscript was to present and compare various (simple) kinetic models. I selected the experimental results of ion transport through PIM, in which typical carriers were used. It is always hard to describe various mathematical models that can be applied to transport description. As I mentioned above, in my manuscript, I wanted to show how to choose an appropriate one. However, I realize that these models are related to the physicochemistry of the transport, but this issue goes beyond the scope of manuscript.
Q11: Example: - Explain why you did not observe a diffusion of Cu(II) ions through the membrane with TOPO as carrier.
RESPONSE: Please see the answer to Q7.
Q12: Specify the operating conditions of the work: temperature, acidity, agitation, etc.
RESPONSE: Some additional information was added:
“Only for the system with D2EHPA as a carrier, due to its properties, a 0.002 M solution of metal ions (initial pH=4) without HCl was used....”
“The aqueous solutions were pumped from external reservoirs by the peristaltic pump (GILSON MINIPULS 3) at a flow rate 16 cm3/min.”
“All experiments were carried out at room temperature (25±2°C).”
Q13: There are no studies to characterize engineered membranes, (SEM/EDS/IFTR)
RESPONSE: The main aim of this manuscript wasn’t to characterize PIMs used in the transport examples but to present and compare various kinetic models. The most important is that the membranes used in the experiments worked stably. No unusual concentration changes were observed during transports, evidence of mechanical damage. There is a lot of literature data about the stability of these membranes containing D2EHPA, Aliquat 336, Cyphos IL 101, and TOPO as carriers. The results presented by the other authors prove the homogeneous structure of these membranes and confirm the long-term stable operation. In the case of membranes containing reactive ionic liquid (RILC8_Br), the respective results are presented in [16].
Reviewer 3 Report
In the presented manuscript the author shows a comparison of mathematical models to describe the transport of metal ions through polymeric inclusion membranes. The studies presented are interesting, but the manuscript must be improved.
Major
1. In experimental section author should precisely describe standard atomic absorption method used for determination of metal ion concentration in the solution. Information about the volume of the sample, if one or more sample were taken for analysis in each time point, the uncertainty of the measurement, if one or both phases were analyzed ect. In cited references all necessary information are not presented.
2. Section 3.1 – influence of the feed and stripping solution volume changes.
Author should more precisely describe the model that was used for the description of the volume changes in the system. Author should present the plot of volume of feeding and receiving phases vs. time together with model which describe it (3rd degree polynominal function). These data can be presented in supplementary file.
3. Author should present concentration vs. time plots for all experiments (not only for one for each carrier type in membrane). These data can be presented in supplementary file.
4. Author should present all calculated constants for each model. These data can be presented in supplementary file.
5. It should be pointed that for each system where best-fitted model is model no. 4 – like in case of Pb(II) and Cd(II) ions in table 4, model no. 5 also present the same fit quality (the difference is negligible and mostly connected with precision of calculation) because model 5 should simplify to model no. 4 for k -1=0. When author will present all constant values from different models it will be easier to analyze the results (probably in that case k -1 is close to 0 but not equal 0).
6.
Minor
1. Page 2, line 50 – additional space is introduced between “cf” and “=”
2. Page 2, line 70 – a space is missing between ref. 4 and words.
3. Page 8, line 244 – In case of Cd(II) ion model no. 4 shows the best fit instead of no. 3.
4. Page 8, line 245 – author write that “…best-fitted models lead to much higher initial maximum fluxes…” but after comparison of initial fluxes calculated from different models it could be seen that the fluxes calculated from best-fitted model are only slightly higher or even lover like in case of Zn(II) and model 4 and 5 (best-fitted).
5. All references should be prepared in the accordance to instruction for authors in the jurnal webpage. In some cases full name of the journal is presented in other the abbreviation sometimes in italic and sometimes in normal form. The font size for ref. 16, 17, 32 is different ect. Author should carefully check all references.

Author Response
I would like to thank the referee for evaluating my manuscript. I have tried to address all the reviewers’ concerns in a proper way and believe that my paper has improved considerably.
REV#3
In the presented manuscript the author shows a comparison of mathematical models to describe the transport of metal ions through polymeric inclusion membranes. The studies presented are interesting, but the manuscript must be improved.
Major
Q1: In experimental section author should precisely describe standard atomic absorption method used for determination of metal ion concentration in the solution. Information about the volume of the sample, if one or more sample were taken for analysis in each time point, the uncertainty of the measurement, if one or both phases were analyzed ect. In cited references all necessary information are not presented.
RESPONSE: According to the Reviewer expectations, some information was added:
During the transport experiments, 1 ml samples from the feed and the receiving solution were taken for analysis by the flame atomic absorption method using SPECTRAA 20ABQ Varian spectrophotometer. Atomic absorption spectroscopy operating parameters are presented in a Supplementary file (Table S2). All experiments were carried out at room temperature (25±2°C).
Q2. Section 3.1 – influence of the feed and stripping solution volume changes.
Author should more precisely describe the model that was used for the description of the volume changes in the system. Author should present the plot of volume of feeding and receiving phases vs. time together with model which describe it (3rd degree polynominal function). These data can be presented in supplementary file.
RESPONSE: The example of a respective plot was presented in the supplementary file. In the manuscript, the following explanations were added:
The example of 3rd-degree polynomial fitting to the experimental results is presented in the Supplementary file (Figure S1).
Q3.Author should present concentration vs. time plots for all experiments (not only for one for each carrier type in membrane). These data can be presented in supplementary file.
RESPONSE: The best-fitted models for all of the experiments and ions were presented in the Supplementary file.
Q4: Author should present all calculated constants for each model. These data can be presented in Supplementary file.
RESPONSE: All calculated pseudo-kinetic constants and initial maximum fluxes were presented in the Supplementary file.
Q5.It should be pointed that for each system where best-fitted model is model no. 4 – like in case of Pb(II) and Cd(II) ions in table 4, model no. 5 also present the same fit quality (the difference is negligible and mostly connected with precision of calculation) because model 5 should simplify to model no. 4 for k -1=0. When author will present all constant values from different models it will be easier to analyze the results (probably in that case k -1 is close to 0 but not equal 0).
RESPONSE: All the calculated pseudo-kinetic constants are presented in the supplementary file. However, this does not change the method and results for selecting the best model.
Minor
Q6: Page 2, line 50 – additional space is introduced between “cf” and “=”
RESPONSE: Done
Q7: Page 2, line 70 – a space is missing between ref. 4 and words.
RESPONSE: “calixresorcin[4]arene derivatives” are substances used as a carrier.
Q8: Page 8, line 244 – In case of Cd(II) ion model no. 4 shows the best fit instead of no. 3.
RESPONSE: Of course, Model No.4 shows the best fit. Thank you. Due to the renumbering of models in the reviewed manuscript, this number has remained the same.
Q9: Page 8, line 245 – author write that “…best-fitted models lead to much higher initial maximum fluxes…” but after comparison of initial fluxes calculated from different models it could be seen that the fluxes calculated from best-fitted model are only slightly higher or even lover like in case of Zn(II) and model 4 and 5 (best-fitted).
RESPONSE: I changed the respective sentence into:
In this membrane system, the best–fitted models lead to initial maximum fluxes higher from 7 to 11%, especially for preferentially transported Cd(II) and Zn(II) ions.
Which I think is more appropriate.
Q10: All references should be prepared in the accordance to instruction for authors in the jurnal webpage. In some cases full name of the journal is presented in other the abbreviation sometimes in italic and sometimes in normal form. The font size for ref. 16, 17, 32 is different ect. Author should carefully check all references.
RESPONSE: All references were checked and corrected according to the Journal editor's expectations.
Round 2
Reviewer 2 Report
The results are already published and in addition there is a contradiction between the published results (https://doi.org/10.1016/j.memsci.2021.119674) and the submitted results.
The manuscript has a serious problem of plagiarism, You must clearly show the difference and the novelty compared to your results already published
The author's answers are not convincing
Author Response
English language was checked by a native speaker.
Q1: The results are already published and in addition there is a contradiction between the published results (https://doi.org/10.1016/j.memsci.2021.119674) and the submitted results.
The manuscript has a serious problem of plagiarism, You must clearly show the difference and the novelty compared to your results already published
The author's answers are not convincing
RESPONSE:
As I previously stated, the experimental results presented in this manuscript are new and have not been published anywhere. The results presented in my previous manuscript are different. In the case of RILC8_Br as a carrier, the main difference is the thickness of the membranes. In a previous manuscript it was (0.096 ± 0.015) mm whereas in this manuscript I applied membrane of thickness equal to 0.0554(±0.0024). While checking the manuscript I found a mistake in Table 1, because in the case of RILC8_Br, the membrane composition was as follow: 50 wt.% of a carrier, 32 wt.% of NPOE, and 19 wt.% of CTA. However, to fulfill the Reviewer expectations, I decided to change these results to the results in which the membrane composition was as follow: 50 wt.% of RILC8_Br, 20 wt.% of NPOE and 30 wt.% of CTA. The calculated new results are presented in Tab. 6, Fig. 6, and Supplementary file (Fig.S6). Note that the model No.2 is still the best fitted one for the new experimental results.
The main aim of this manuscript was to present and compare various (simple) kinetic models. In the manuscript I used typical results of ion transport through PIMs, because I wanted to show that not only the Danesi model can be used. A novelty is the model selection method presented in the paper, based on appropriate fit quality parameters and runs test. I also found that models which never were used for transport description through PIMs (Models No.3 and 4), can be successfully applied and lead to a much better fit.
Taking into account the Reviewer#2 and #3 expectations, I modified my manuscript and add some additional explanations and references:
In the introduction section:
Despite of many different kinetic models that enable the determination of the initial maximum flux (JM) and a quantitative description of the concentration change in the feed and receiving solutions, some authors calculating JM from the first derivative of an exponential decay function fitted to the feed solution concentration dependence [33-37]. This approach probably results from problems occurring when fitting the most commonly used kinetic model (model No. 2) to the experimental data. The calculated JM value describes the system efficiency. However it is impossible to obtain a quantitative description of time-dependent concentrations especially in the receiving solution. In this case, there is also no possibility of a physicochemical interpretation of the exponential decay function, contrary to the typical kinetic models.
[33] Hoque, B.; Almeida, M.I.G.S.; Cattrall, R.W.; Gopakumar, T.G.; Kolev, S.D. Effect of cross-linking on the performance of polymer inclusion membranes (PIMs) for the extraction, transport and separation of Zn(II). J. Membr. Sci. 2019, 589, 117256, DOI: 10.1016/j.memsci.2019.117256.
[34] St John, A.M.; Cattrall, R.W.; Kolev, S.D. Determination of the initial flux of polymer, inclusion membranes. Separ. Purif. Technol. 2013, 116, 41–45, DOI: 10.1016/j.seppur.2013.05.021.
[35] O’Bryan, Y.; Cattrall, R.W.; Truong, Y.B.; Kyratzis, I.L.; Kolev, S.D. The use of poly(vinylidenefluoride-co-hexafluoropropylene) for the preparation of polymer inclusion membranes. application to the extraction of thiocyanate. J. Membr. Sci. 2016, 510, 481-488, DOI: 10.1016/j.memsci.2016.03.026.
[36]Croft, Ch.F.; Almeida, M.I.G.S.; Cattrall, R.W.; Kolev, S.D. Separation of lanthanum(III), gadolinium(III) and ytterbium(III) from sulfuric acid solutions by using a polymer inclusion membrane, J. Membr. Sci. 2018, 545, 259–265, DOI: 10.1016/j.memsci.2017.09.085.
[37] Bonggotgetsakul, Y.Y.N.; Cattrall, R.W.; Kolev, S.D. The Effect of Surface Confined Gold Nanoparticles in Blocking the Extraction of Nitrate by PVC-Based Polymer Inclusion Membranes Containing Aliquat 336 as the Carrier. Membranes 2018, 8, 6, doi:10.3390/membranes8010006.
and:
The main aim of this work is, therefore, to discuss the applicability of the above simple kinetic models and to select the one as more appropriate for the transport description of substances through PIMs. The usefulness of the models was evaluated with the use of typical parameters describing the model–fit quality to the experimental results of Zn(II), Cd(II), Pb(II), and Cu(II) ions transport. In model calculations, a change in the feed and stripping solution volume (because of sampling) was taken into account. The goodness–of–fit evaluation was estimated by the standard error of the regression, Akaike information criterion (AIC), Bayesian (Schwarz) information criterion (BIC), and Hannan-Quinn information criterion (HQC). A nonparametric runs test was also used to examine the randomnessof residuals.
The novelty of this research is the development of the method for selection of a proper model based on appropriate fit quality parameters and runs test as well as the application of kinetic models, which have never been used for transport description through PIMs.
In Conclusions section
An appropriate mathematical model enables a quantitative prediction of concentration changes in external solutions and calculation of e.g. the time needed to achieve maximum concentration, recovery, or enrichment factor of transported substances.
Among the numerous equations used to describe transport kinetics, it is impossible to choose only one as most appropriate. The presented results indicated that the application of the most frequently used model (proposed by Danesi) is significantly limited because of nonrandomly distributed residuals.
The selection of the appropriate model should be based on the values of the standard error of the regression, Akaike Information Criterion (AIC), Bayesian (Schwarz) Information Criterion (BIC), or Hannan-Quinn Information Criterion (HQC) after the runs test evaluation (residuals randomness check).
From the models presented in this report, the model No. 4 is most universal. However, the model selection should be individual for each experimental relationship. It was also found that a non–linear equation (3rd degree polynomial) can be successfully used to describe solutions volume changes in a membrane system (because of sampling) and leads to a better fit of a model to experimental data.
The results indicated that the models that have not been used so far for transport description in PIMs, i.e. the models No. 3 and 4, can be successfully applied. These models are particularly important in the case of systems where time-lag is observed. This mainly applies to the systems with carriers characterized by a high partition coefficient (high sorption of transported substances to the membrane), slow diffusion inside the membrane (e.g. because of relative high membrane thickness) or slow kinetics of extraction and reextraction at the respective membrane interfaces.
All of changes in my text are automatically display (track changes option on).
Reviewer 3 Report
In part of the manuscript author use abbreviation "Tab." (line 154, 223, 305, 325) in the rest part author use full name "Table". It should be unified.
Author Response
REV#3
According to Reviewer #3 expectations, the introduction, references, and conclusions were I hope improved.
For example in the introduction section additional explanation and references were added:
Despite of many different kinetic models that enable the determination of the initial maximum flux (JM) and a quantitative description of the concentration change in the feed and receiving solutions, some authors calculating JM from the first derivative of an exponential decay function fitted to the feed solution concentration dependence [33-37]. This approach probably results from problems occurring when fitting the most commonly used kinetic model (model No. 2) to the experimental data. The calculated JM value describes the system efficiency. However it is impossible to obtain a quantitative description of time-dependent concentrations especially in the receiving solution. In this case, there is also no possibility of a physicochemical interpretation of the exponential decay function, contrary to the typical kinetic models.
[33] Hoque, B.; Almeida, M.I.G.S.; Cattrall, R.W.; Gopakumar, T.G.; Kolev, S.D. Effect of cross-linking on the performance of polymer inclusion membranes (PIMs) for the extraction, transport and separation of Zn(II). J. Membr. Sci. 2019, 589, 117256, DOI: 10.1016/j.memsci.2019.117256.
[34] St John, A.M.; Cattrall, R.W.; Kolev, S.D. Determination of the initial flux of polymer, inclusion membranes. Separ. Purif. Technol. 2013, 116, 41–45, DOI: 10.1016/j.seppur.2013.05.021.
[35] O’Bryan, Y.; Cattrall, R.W.; Truong, Y.B.; Kyratzis, I.L.; Kolev, S.D. The use of poly(vinylidenefluoride-co-hexafluoropropylene) for the preparation of polymer inclusion membranes. application to the extraction of thiocyanate. J. Membr. Sci. 2016, 510, 481-488, DOI: 10.1016/j.memsci.2016.03.026.
[36]Croft, Ch.F.; Almeida, M.I.G.S.; Cattrall, R.W.; Kolev, S.D. Separation of lanthanum(III), gadolinium(III) and ytterbium(III) from sulfuric acid solutions by using a polymer inclusion membrane, J. Membr. Sci. 2018, 545, 259–265, DOI: 10.1016/j.memsci.2017.09.085.
[37] Bonggotgetsakul, Y.Y.N.; Cattrall, R.W.; Kolev, S.D. The Effect of Surface Confined Gold Nanoparticles in Blocking the Extraction of Nitrate by PVC-Based Polymer Inclusion Membranes Containing Aliquat 336 as the Carrier. Membranes 2018, 8, 6, doi:10.3390/membranes8010006.
and:
The main aim of this work is, therefore, to discuss the applicability of the above simple kinetic models and to select the one as more appropriate for the transport description of substances through PIMs. The usefulness of the models was evaluated with the use of typical parameters describing the model–fit quality to the experimental results of Zn(II), Cd(II), Pb(II), and Cu(II) ions transport. In model calculations, a change in the feed and stripping solution volume (because of sampling) was taken into account. The goodness–of–fit evaluation was estimated by the standard error of the regression, Akaike information criterion (AIC), Bayesian (Schwarz) information criterion (BIC), and Hannan-Quinn information criterion (HQC). A nonparametric runs test was also used to examine the randomnessof residuals.
The novelty of this research is the development of the method for selection of a proper model based on appropriate fit quality parameters and runs test as well as the application of kinetic models, which have never been used for transport description through PIMs.
As well as in Conclusions section:
An appropriate mathematical model enables a quantitative prediction of concentration changes in external solutions and calculation of e.g. the time needed to achieve maximum concentration, recovery, or enrichment factor of transported substances.
Among the numerous equations used to describe transport kinetics, it is impossible to choose only one as most appropriate. The presented results indicated that the application of the most frequently used model (proposed by Danesi) is significantly limited because of nonrandomly distributed residuals.
The selection of the appropriate model should be based on the values of the standard error of the regression, Akaike Information Criterion (AIC), Bayesian (Schwarz) Information Criterion (BIC), or Hannan-Quinn Information Criterion (HQC) after the runs test evaluation (residuals randomness check).
From the models presented in this report, the model No. 4 is most universal. However, the model selection should be individual for each experimental relationship. It was also found that a non–linear equation (3rd degree polynomial) can be successfully used to describe solutions volume changes in a membrane system (because of sampling) and leads to a better fit of a model to experimental data.
The results indicated that the models that have not been used so far for transport description in PIMs, i.e. the models No. 3 and 4, can be successfully applied. These models are particularly important in the case of systems where time-lag is observed. This mainly applies to the systems with carriers characterized by a high partition coefficient (high sorption of transported substances to the membrane), slow diffusion inside the membrane (e.g. because of relative high membrane thickness) or slow kinetics of extraction and reextraction at the respective membrane interfaces.
All of changes in my text are automatically display (track changes option on).
Q1: In part of the manuscript author use abbreviation "Tab." (line 154, 223, 305, 325) in the rest part author use full name "Table". It should be unified.
RESPONSE: Done.